# Ionospheric Upwelling and the Level of Associated Noise at Solar Minimum

Timothy Wemimo David[1,2], Chizurumoke Michael Michael[3,2], Darren Wright[2], Adetoro Temitope Talabi[1], and Abayomi Ekundayo Ajetunmobi[1]

[1]Department of Physics, Olabisi Onabanjo University, Ago-Iwoye, Nigeria
[2]Department of Physics and Astronomy, University of Leicester, Leicester, UK
[3]Faculty of Arts, Science and Technology (FAST), University of Northampton, Northampton, UK

**Correspondence:** Chizurumoke Michael Michael (michael.michael@northampton.ac.uk)

**Abstract.** We have studied the ionospheric upwelling with magnitude above $10^{13} m^{-2} s^{-1}$ using the data during the IPY-ESR 2007 campaign, which coincides with solar minimum. The noise level in low, medium and high-flux upflows is investigated. We found that the noise level in high-flux upflow is about 93% while the low and medium categories are 62% and 80%, respectively. This shows that robust and stringent filtering techniques must be ensured when analysing incoherent data in order

not to bias the result. Analysis reveals that the frequency of the low-flux upflow events is about 8 and 73 times the medium and high-flux upflow events, respectively. Seasonal observation shows that the noise level in the upflow classes is predominantly during winter. The noise is minimal in summer, with a notable result indicating occurrence of actual data above noise in the low-flux class. Moreover, the percentage occurrence of the noise level in the data increases with increasing flux strength, irrespective of the season. Further analysis reveals that the noise level in the local time variation peaked around 17 – 18 LT and

minimum around local noon.

## 1 Introduction

The European Incoherent Scatter Scientific Association (EISCAT, Rishbeth, 1985) an international scientific body set up to carry out research, using the Incoherent Scatter Radar (ISR) technique to probe the ionosphere as well as the different layers of the atmosphere (Fu et al., 2015). The electromagnetic pulses transmitted from the radar interact with the ionospheric plasma,

and the latter emits a fractional part of the exploring signal as scattering. The backscatter frequency spectrum received (referred to as the incoherent scatter (IS) spectrum) provides various information on properties and state of the ionosphere (Rishbeth and Williams, 1985). Several key ionospheric parameters can be derived from the IS spectrum (e.g., Gordon, 1958; Dougherty and Farley, 1961; Evans, 1969; Alcayde, 1997; Li et al., 2012). Such parameters include the electron density, ion and electron temperature, and the ion drift velocity relative to the radar.

The data from ISR have been previously analysed by several authors. For example, Ogawa et al. (2009) used the ISR data to show that ionospheric upwelling can occur at any local time (LT). Vlasov et al. (2011), while analysing the EISCAT data of international polar year (IPY) 2007 have shown that travelling ionospheric disturbances and atmospheric gravity waves are common high-latitudes phenomena, and frequent during local summer. More recently, David et al. (2018), using the same set

of data, have shown that the maximum occurrence peak of ion upwelling, irrespective of the class, occurs around local noon.

Such analysis of data from Tromsø, where EISCAT VHF radar operates, shows that ionospheric upflow and downflow are possible under any level of geomagnetic condition (Endo et al., 2000). According to Foster et al. (1998) in their study of some of the EISCAT frequently run programmes, occurrence frequency of upwelling ions has a direct relationship with increase in geodetic altitude. The study of stimulated electromagnetic emission by EISCAT heating facility at Tromsø, used to modulate the ionosphere for experimental purposes, has shown that reduction is observed in the elevated electron temperature when the

radio pumping is close to the gyro-harmonic frequency of the electron (Fu et al., 2015). Williams (1995) in his analysis of the initial phase of the EISCAT Svalbard Radar (ESR) observation proposed that to properly investigate the polar ionosphere dynamics, a facility that will address the **k** vector at a time (3-antenna facility) should be considered instead of the usual method of a single antenna swinging through the $x-$, $y-$ and $z-$directions in sequence. Although the ESR facility like other IS radars is built with high gain and low noise performance owing to its transmitted power (up to a maximum of 1.0 MW),

antenna sensitivity (42 m diameter) and high latitude location ($78°09'11''N$), there are noise from other sources such as the signal-to-noise ratio (SNR) that varies inversely as the square of the distance from the receiver to the target (i.e.,$S \approx R^{-2}$), noise associated with clutter in altitude up to 140 km (Wannberg et al., 1997) and the electromagnetic noise at the background. Lehtinen (1989) and Vierinen et al. (2008) have suggested that the accuracy of the autocorrelation function in radar backscatter is limited as a result of disturbances from noise. David et al. (2018) worked on the technique to filter the real data from noise,

but no statistical analysis to quantify the level of noise was carried out. Li et al. (2020) in their attempt to simulate the SNR of a proposed ISR (phased array radar) and compared with an equivalent parabolic dish radar, showed theoretically through their findings that the SNR from the phased array radar is weaker compared to that of the equivalent parabolic dish, whereas the analysis of noise and its error were left for future work.

In order to avoid radar data that are susceptible to clutter as a result of mountainous topography of Svalbard (David et al.,

2018), the data analysed in this work were observed by the EISCAT Svalbard Radar (ESR) 42 m dish between the altitude range of 100 km and 470 km (where noise associated to clutter and background electromagnetic effect have been filtered) with a time resolution of 1 minute. As such, the focus of this paper is the analysis of the statistical occurrence of noise associated with different classes of ionospheric upflow, local time (LT) dependence, as well as seasonal variability of the noise during ESR observations of upwelling ions at solar minimum of 2007 – 2008 shown in Figure 1, where the maximum daily total sunspot

number is 66.0 in 2007 and 60.0 in 2008.

Such statistical studies have potential application in the improvement of the EISCAT instrumentation. For example, in the development of the upgrade of the existing EISCAT radars, the EISCAT 3D. This is because, for example, noise from sources such as the signal-to-noise ratio influence the temporal resolution of the EISCAT 3D radar measurements (Stamm et al., 2021). The EISCAT 3D radar relies on a high-power and phased array system can produce three-dimensional imaging of the upper

atmospheric structures and processes in high resolution (McCrea et al., 2015). With such high-resolution imaging capabilities of the EISCAT 3D radar data, they can enhance research in, for instance, ionospheric electron densities and ion flow velocities. Thus, the present study can contribute to the development of the recent EISCAT 3D radar.

## 2 Instrumentation and Data

The primary data used for this work is sourced from EISCAT Svalbard radar (ESR) during the international polar year (IPY) campaign in 2007. The ESR is a fixed and field-aligned 42m dish. Basic ionospheric parameters measured by the ESR are the electron density, electron and ion temperature and, the ion velocity which are respectively abbreviated as: $n_e$, $T_e$, $T_i$, and $v_i$. In addition, about 300 days observation of 312,444 field-aligned profiles was made and the observation occurs during a deep solar minimum as shown in Figure 1.

The ESR observations of upwelling ions at solar minimum of 2007 – 2008 shown in Figure 1, indicates that the maximum daily total sunspot number is 66.0 in 2007 and 60.0 in 2008. Likewise, the maximum daily F10.7 radio flux over the same period as shown in Figure 1 is 93.9 and 88.6 in 2007 and 2008 respectively. Noise or rejected data in this study refers to ISR data with very high values of unphysical velocities above 10 km s$^{-1}$ unintentionally obtained during incoherent scatter analysis (Jones et al., 1988; Blelly et al., 1996; David et al., 2018). The classes of flux ($\geq 7.5 \times 10^{13}$ $m^{-2} s^{-1}$; Wahlund and Opgenoorth (1989)) in this study and the filtering methodology follow the work by David et al. (2018), where upflows are categorised as follows:

- Low-flux upflow, $1.0 \times 10^{13} m^{-2} s^{-1} \leq f_{ion} < 2.5 \times 10^{13} m^{-2} s^{-1}$]

- Medium-flux upflow, $2.5 \times 10^{13} m^{-2} s^{-1} \leq f_{ion} < 7.5 \times 10^{13} m^{-2} s^{-1}$

- High-flux upflow, $f_{ion} \geq 7.5 \times 10^{13} m^{-2} s^{-1}$

## 3 Results and Discussion

Figures 2 and 3 show the EISCAT Svalbard Radar ionospheric parameters plot for a dayside plot on August 12, 2007 and a nightside plot on December 28, 2007 respectively. The dayside plot in August 12, 2007 shows when the data is less noisy, while the nightside event represents a typical example of periods when ISR data is enmeshed with random unwanted data without physical meaning. The panels (a) to (e) on both figures are respectively the electron density, electron temperature, ion temperature, ion drift velocity, and the ion flux. August 12, 2007 plot indicates intermittent moderately intense electron precipitation down to the E region from 06:00 UT to around 10:30 UT, and thereafter remains predominantly moderate throughout the rest of the period. On the other hand, the nightside event of December 28, 2007 shows that the ionosphere was predominantly quiet with little or no electron precipitation to the E region, expect for the evening time. The F region electron density in Figure 2 indicates a long duration of elevation whereas, the F region electron density did not record significant enhancement in Figure 3. On the second topmost panel is the electron temperature, which indicates corresponding enhancement to the period of precipitation during the August 12, 2007 event shown in Figure 2, while the same panel on Figure 3 shows a noisy period especially at the lower and higher altitude. The middle panel indicates that a moderate with few intermittent intense ion temperature dominate the period on August 12, 2007 event. The period between 22:00UT on December 28, 2007 and 02:00 UT the following day, indicates a mixture of moderate and elevated ion temperature. Panels d and e on Figure 2 show accelerated ions

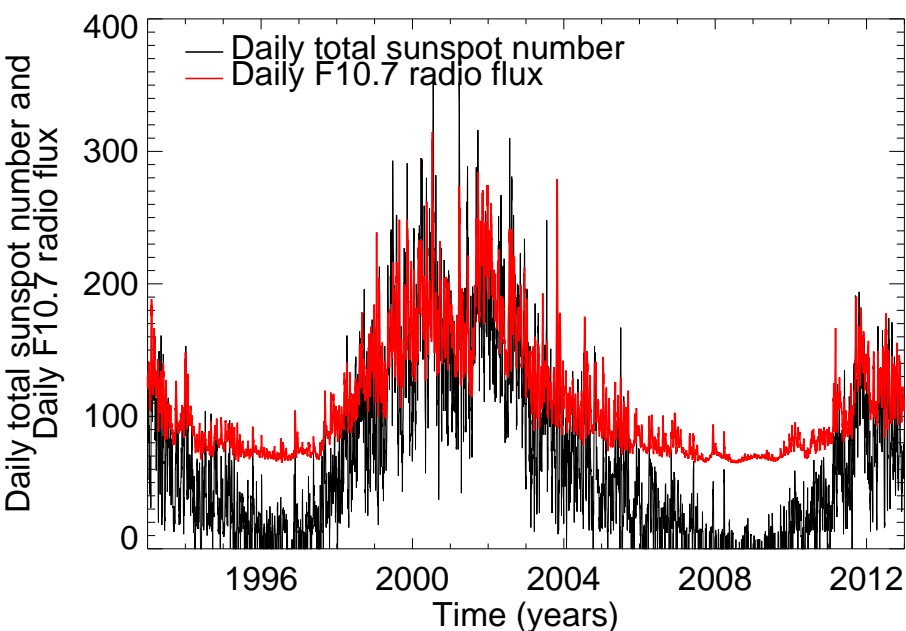

**Figure 1.** Daily sunspot number indicated by black and daily F10.7 radio flux indicated by red line adapted from David et al. (2018).

around afternoon and a corresponding high-flux respectively, whereas the ion velocity in Figure 3 are unphysical and there is no corresponding high-flux upflow.

Table 1 shows the number of data points for both unfiltered and actual (filtered) data for each class of ion upflow flux, as well as the percentages of the actual and noisy data. The actual data is the number of data points that satisfy the filtering technique (used in this work) set by David et al. (2018) for upwelling ions, while unfiltered data on the other hand is the number of data points before filtering that fell in the range of each class of upflow from the raw data obtained by the ESR during the period under investigation. The percentage of the actual data is calculated from the percentage ratio of the actual to the unfiltered data points. On the other hand, percentage noise for each class is obtained by

$$noise = \left(1 - \frac{actual\ data\ point\ for\ each\ class}{unfiltered\ data\ point\ for\ each\ class}\right)$$

The low-flux upflow is a common event and analysis of filtered data reveals that its frequency is about 8 and 73 times the medium- and high-flux upflow events, respectively. The data in Table 1 indicates that about 33% of the ESR data satisfies the filter, of which about 29% contribution is from the low-flux upflow, while medium- and high-flux upflow contribute about

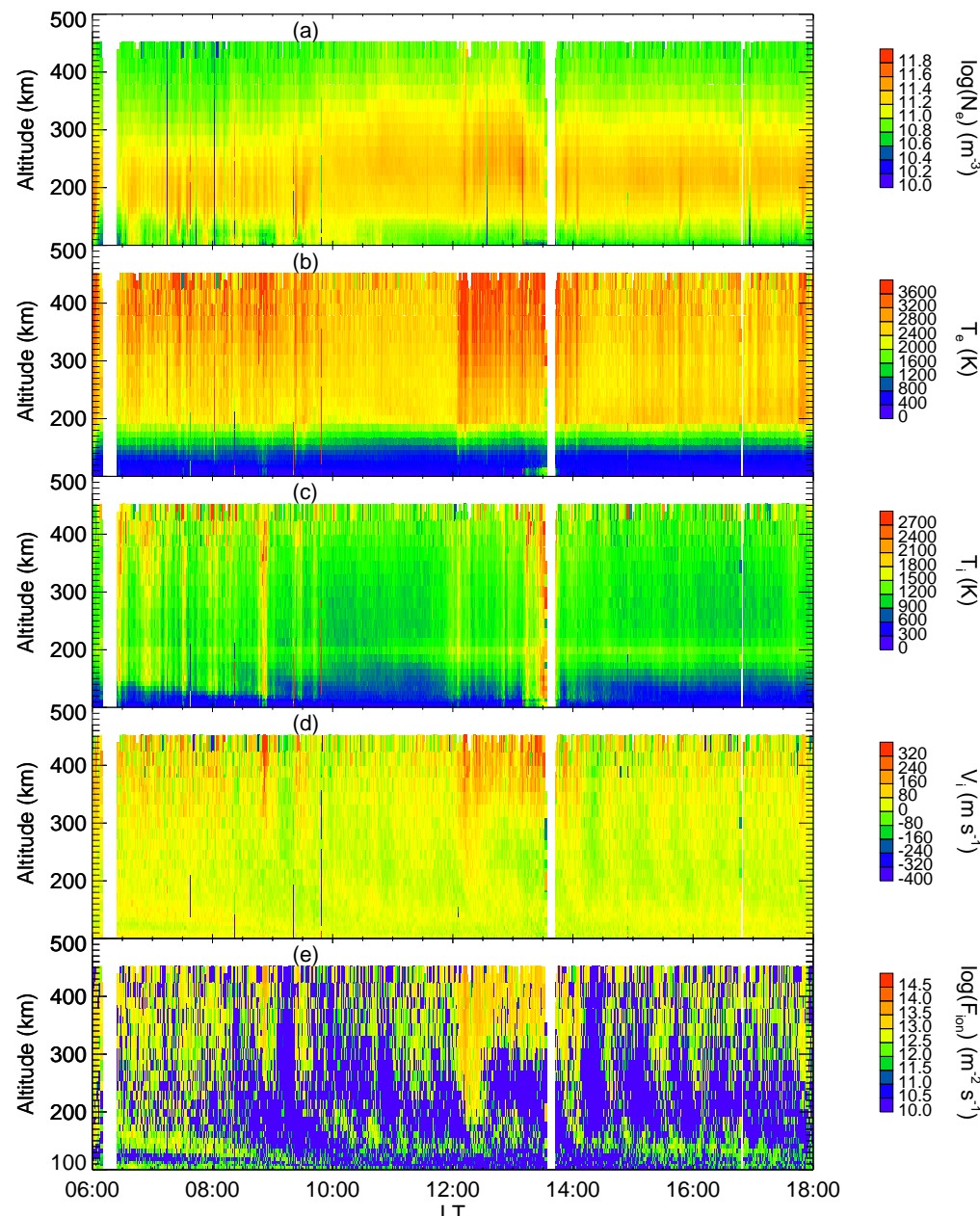

**Figure 2.** EISCAT Svalbard Radar (42 m dish) parameter plot for dayside 11 September, 2007. The panels a – e are respectively the electron density, electron temperature, ion temperature, ion drift velocity and the ion flux.

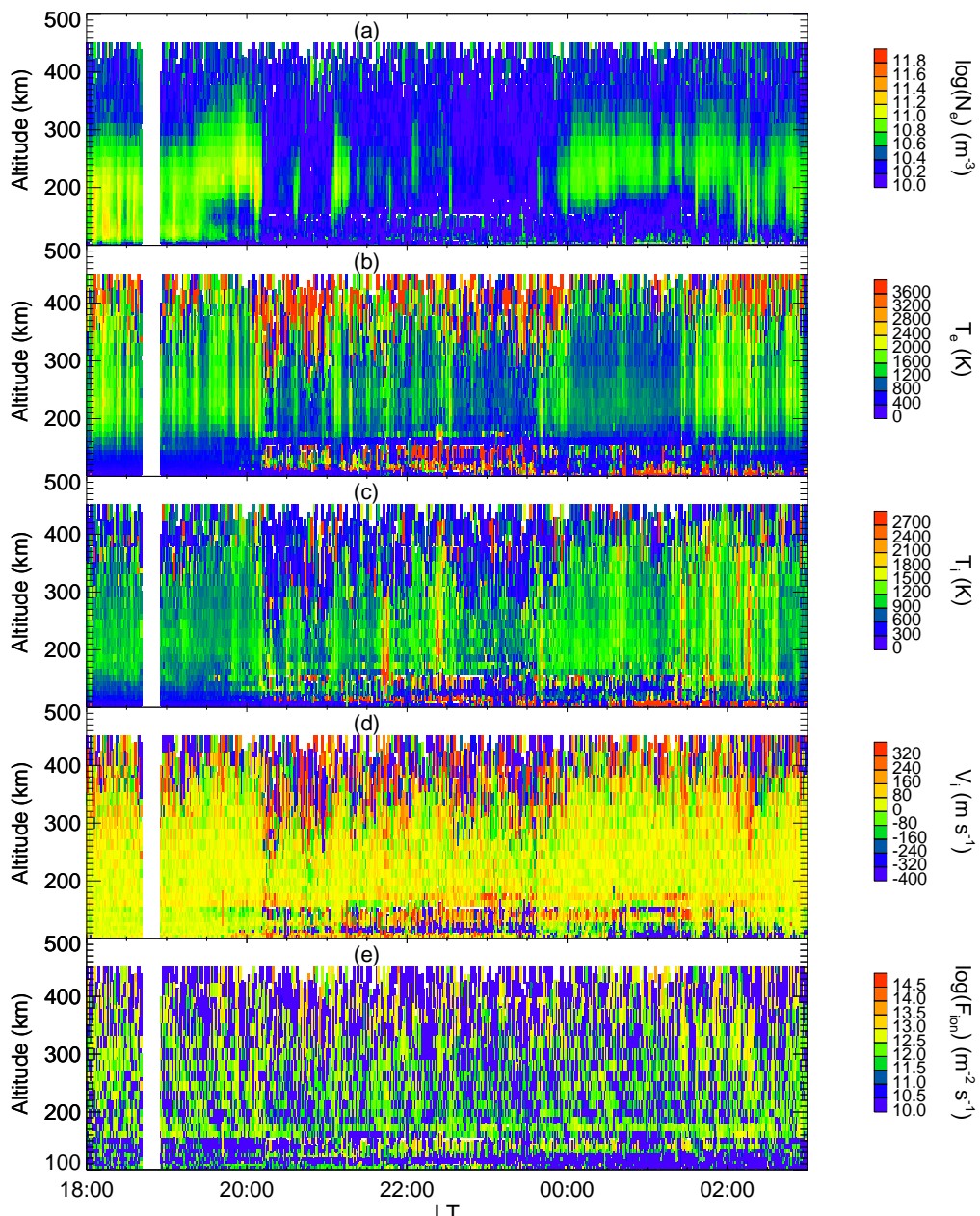

**Figure 3.** EISCAT Svalbard Radar (42 m dish) parameter plot for nightside 28 December, 2007. The panels a – e are respectively the electron density, electron temperature, ion temperature, ion drift velocity and the ion flux.

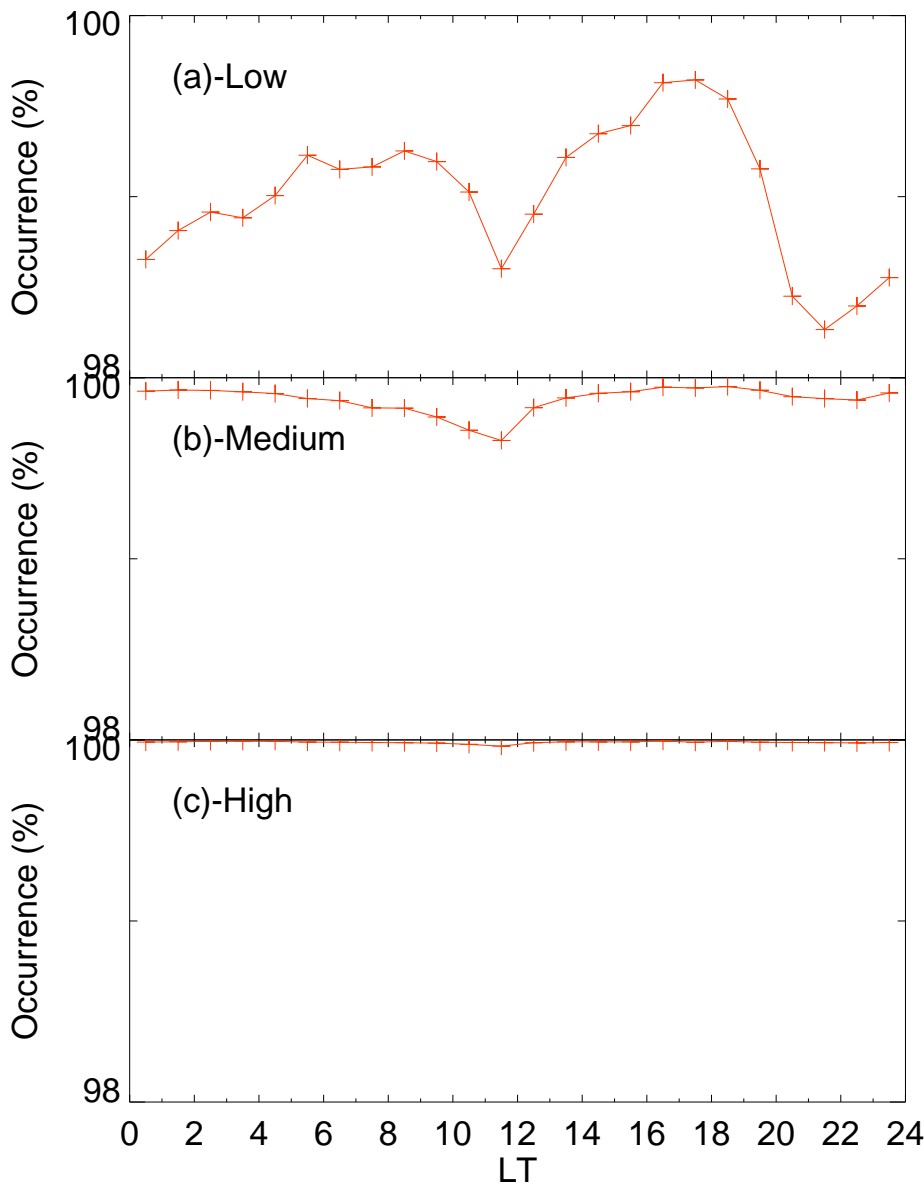

**Figure 4.** Local time distribution of noise occurrence in (a) low, (b) medium, and (c) high flux upflow.

3.4% and 0.4% respectively. Investigation shows that the levels of noise in the three classes of upflow are about 62%, 80% and 93% for the respective low-, medium-, and high-flux upflows. It is well known that ISR data are entangled in noise, the high

level occurrence observed here may be attributed to low signal-to-noise ratio characterising much of the high-latitude data at deep solar minimum around the period (David et al., 2018). It is therefore left opened to research to investigate whether data outside the solar minimum would have a lower rejection rate.

The geographical location of the ESR reported by David et al. (2018) subjects it to variable radiation flux from the Sun and a variable rate of ionisation across the seasons. The result from Table 2, under the column for actual data/noise heading, reveals that the percentage of rejected data increases from low-flux upflow to high-flux upflow irrespective of the season. The noise level in the low-flux upflow ranges from 48 – 80% across the season, whereas the medium- and high-flux upflow respectively as 68 – 87% and 87 – 95%. However, there is no any definite pattern across the season for the upflow classes. Moreover the noise level for each category of upflow is predominantly during winter and this is not unconnected with the radar data being usually of lower quality at winter. The noise as expected is minimal in summer, with a notable result as seen in the low-flux showing occurrence of actual data is about 51.6% while noise is approximately 48.4% – the only case in which actual data is above noise. Further analysis as shown in Figure 4 reveals the distribution of the three classes of ion flux upflows with respect to local time interval. The first panel of Figure 4 shows the local time variation of the noise for the low-flux upflow, where a clear trough is observed around local noon and pre-midnight. The peak of the period when noisy data is observed is shown to be between 17 – 18 LT. The minimum percentage of the noise level across LT is above 98. The middle panel of Figure 4 also indicates that the noise level for the medium-flux, though very high (above 99% across LT), is least around local noon. However, no distinct minimum is observed in the high-flux upflow shown on the last panel of Figure 4, in fact, the noisy data for the class approaches 100% for all local time.

It is worthy to note that the dip in the noise occurrence in Figure 4 is as a result of large ion outflows and an elevated ionisation rate, which are characteristics around the cusp (Welling et al., 2015). The contributory role to the suppression of noise in this sector, as well as the midnight sector may be attributed respectively to the soft electron precipitation, which is characteristic of the abundant F-region ionisation, and the reconnection usually experienced at the night side, leading to substorm.

In addition, it appears that high level of rejected data is evident in ISR data and as a result, robust and stringent filtering techniques must be ensured when analysing incoherent radar data in order not to bias the result. Ogawa et al. (2009) and Endo et al. (2000) have pointed out that radar data are noisy in the topside ionosphere as a result of unphysical velocity inadvertently obtained coupled with thermal noise from receivers as well as uncertainties arising from fitting line-of sight velocity.

In the light of the above, the proposed phased array ISR, named Sanya ISR should take into cognisance, an ISR that in practice, will have a better SNR by ensuring the best input radar system constants, effectual scattering volume, and spatial variability terms in space, as stated in the work of Li et al. (2020). The results of this work could also be integrated in the buildup of the EISCAT 3D to allow for comparison in the SNR of the Scandinavian Arctic infrastructure and the Sanya ISR, which is proposed to be the first multistatic ISR in a low latitude region.

## 4  Summary and Conclusions

Noise associated with Incoherent Scatter radar (ISR) data located at Longyearbyen in Svalbard has been investigated during the solar minimum and the results are summarised thus:

– About 33% of the raw data satisfies the filter, of which about 29% contribution is from the low-flux upflow, while medium- and high-flux upflow contribute about 3.4% and 0.4% respectively.

– Investigation shows that the levels of noise in the three classes of upflow are about 62%, 80% and 93% for the respective low-, medium-, and high-flux upflows.

– The percentage occurrence of the noise level in the data increases with increasing flux strength, irrespective of the season.

– The noise level for each category of upflow is predominantly during winter and minimal in summer.

– A notable result as seen in the low-flux during summer shows occurrence of actual data is about 51.6% while noise is approximately 48.4% – the only case in which actual data exceeds noise.

– Local time variation indicates that the noise level peaked around 17 – 18 LT and minimum around local noon.

*Data availability.*  The data used in this paper can be obtained via the schedule pages of the EISCAT website (https://www.eiscat.se/) and the Madrigal database.

*Author contributions.*  TWD; conceptualisation, methodology, formal analysis, writing-original draft, project administration, C.M.M; methodology, writing-review and editing, validation, project administration, DMW; Conceptualization, supervision, resources, ATT; writing-review and editing, project administration, AEA; writing-review and editing, project administration. All authors have read and agreed to the published version of the manuscript.

*Competing interests.*  The authors declare no conflict of interest.

*Acknowledgements.*  The authors show their appreciation to the EISCAT Scientific Association for easy accessibility to the Madrigal database. The international community is equal applauded for the 2007 international polar year campaign that generated such high level of robust data. Great thanks to Tertiary Education Trust Fund (TETFund) of Nigeria and the Olabisi Onabanjo University, Ago-Iwoye, Nigeria, for sponsoring this research. The University of Leicester, United Kingdom, is acknowledged for allowing the use of UoL Spectre in the analysis.

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

**Table 1.** Ion flux classification with associated signal and noise ratio.

| Ion Flux | Unfiltered data point | Actual data point | Actual data (%) | Noise (%) |
|---|---|---|---|---|
| Low | 219772 | 82852 | 37.70 | 62.30 |
| Medium | 49650 | 9810 | 19.76 | 80.24 |
| High | 16018 | 1131 | 7.06 | 92.94 |

**Table 2.** Seasonal variation of the signal and noise ratio for different classes of ion flux upflow.

| Ion Flux | Number of unfiltered data **Number of actual (filtered) data** | | | | Accepted data (%) **Rejected data (%)** | | | |
|---|---|---|---|---|---|---|---|---|
| | spring | summer | autumn | winter | spring | summer | autumn | winter |
| Low | 69847 (**23651**) | 75154 (**38754**) | 30619 (**11549**) | 44152 (**8898**) | 33.86 (**66.14**) | 51.57 (**48.43**) | 37.72 (**62.28**) | 20.15 (**79.85**) |
| Medium | 13180 (**2794**) | 7583 (**2411**) | 7382 (**1640**) | 21505 (**2965**) | 21.20 (**78.80**) | 31.79 (**68.21**) | 22.22 (**77.78**) | 13.79 (**86.21**) |
| High | 2665 (**171**) | 1866 (**225**) | 2196 (**122**) | 9291 (**613**) | 6.42 (**93.58**) | 12.06 (**87.94**) | 5.56 (**94.44**) | 6.60 (**93.40**) |