# Peer review of "Ionospheric Upwelling and the Level of Associated Noise at Solar Minimum"

_Annales Geophysicae, 2023_

## Author Comment (AC1)

**Ionospheric Upwelling and the Level of Associated Noise at Solar Minimum**

Timothy Wemimo David, Chizurumoke Michael Michael, Darren M. Wright, Adetoro Temitope Talabi, and Abayomi E. Ajetunmobi

**Status**: open (until 03 Apr 2024)

**Comment types**: **AC** – author | **RC** – referee | **CC** – community | **EC** – editor | **CEC** – chief editor | : Report abuse
Post a comment   Subscribe to comment alert

- **RC1**: 'Comment on angeo-2023-37', Anonymous Referee #1, 26 Feb 2024   reply

  **Review of "Ionospheric Upwelling and the Level of Associated Noise at Solar Minimum" by David et al.**

  The authors present a statistical study into the occurrence and characteristics of noise in incoherent scatter radar observations. Noise frequently accompanies ISR altitude profiles, and, as researchers in the field well know, one must be aware of its characteristics to properly interpret ISR data. The work is a timely scrutiny of measurement noise based on a large dataset of ISR data. The authors provide a well written introduction to ISR operation and explain how noise is defined. The results are presented in a structured concise manner. I quite enjoyed reading this paper.

  I believe that publishing the paper *as is* is justifiable. Nevertheless, there are potential points of improvement that bear promise should the authors wish to pursue them. The below list is meant to provide some ideas, and I do not expect or demand that the authors pursue all of these points.

  o The authors are rather vague on the mechanisms that produce noise to begin with. Are low ionization levels alone enough to cause local winter noise to dominate in this way? Does noise always appear with the same essential characteristics or is it possible to discern certain physical traits in the noise?

  o How does the noise proportion during local winter respond to the onset of geomagnetic storms? A simple superposed epoch analysis of storm-time onsets (or the onset of other geomagnetic index-excursions) which may show when or whether the signal rises above the noise during such events.

  o The radar in question is well positioned to observe the cusp, where ion outflows are highly characteristic, as well as elevated ionization rates. This sector, as well as the midnight sector, see a dip in the noise occurrence in Figure 4. Notably, the soft electron ionization that is characteristic for the cusp provides abundant F-region ionization. A short discussion of why noise is suppressed in these sectors may be enlightening.

  Sentence 20: "(...) common high-latitude phenomena, and are frequent during local summer."

Sentence 25: should it read "(...) altitudinal increase of the ionosphere"?

Sentence 80: Perhaps the authors can offer preliminary suggestions as to whether and/or why a deep solar minimum is associated with increased levels of noise.

**RESPONSE**

The authors deeply appreciate the reviewer for taking time to painstakingly go through the manuscript. Although the reviewer did not make the minor corrections mandatory, the authors have made reviews as highlighted below following the valuable recommendations from the reviewer on how to improve the manuscript.

- Sentence from line 95 has the following added to it.

It is worthy to note that the dip in the noise occurrence in Figure 4 is as a result of large ion outflows and an elevated ionization rate, which are characteristics around the cusp. The contributory role to the suppression of noise in this sector, as well as the midnight sector may be attributed respectively to the soft electron precipitation, which is characteristic of the abundant F-region ionization, and the reconnection usually experienced at the night side, leading to substorm.

Thank you for suggesting ways of improving the analysis presented in this manuscript. A robust study on geomagnetic storms and noise proportion will be looked at in detail in future work.

- Sentence in line 20: has been edited as suggested by the reviewer as follows:

...common high-latitudes phenomena, and frequent during local summer.

- Sentence in line 25 has been rewritten as follows:

...occurrence frequency of upwelling ions has a direct relationship with increase in geodetic altitude

- A preliminary suggestion to Sentence 80 has been added. The literature cited would be useful for readers.

...occurrence observed here may be attributed to low signal-to-noise ratio characterizing much of the high-latitude data at deep solar minimum around the period (David et al., 2018).

---

## Author Comment (AC2)

**RESPONSE TO REFEREE #2 COMMENTS**

**RC3**: ['Comment on angeo-2023-37'](), Anonymous Referee #2, 08 Apr 2024  reply
This paper examines and classifies the variability of noise occurrence in the ISR ion velocities data. This is useful work that is potentially worth publishing. However, the present version needs a major revision because of rather poor structuring and lack of necessary information mainly related to the introduction and discussion.

Thank you for the positive comments on the importance of this study and for providing valuable recommendations on how to improve the paper as well as clarify confusing text. Please find responses to the points raised below. The responses made to the points raised by the reviewer are written in italics. Those coloured in red are the new input.

Comments.

Ll. 39-40. "The main focus of this paper …" This basic statement does not seem to adequately reflect what is actually being done. First, the statistical occurrence of noise is studied, rather than noise in terms of its inherent properties. Secondly, not only seasonal variability is presented, but also the dependence on LT. Please formulate your goal more precisely.

*The main focus of this paper is the analysis of the statistical occurrence of noise associated with different classes of ionospheric upflow, local time (LT) dependence, as well as seasonal variability of the noise during ESR observations of upwelling ions at solar minimum of 2007 – 2008 shown in Figure 1*

It is also not clear how the present study is placed into context. The sentence preceding "the main focus", with reference to earlier work by Wannberg et al. (1997), lists possible sources of the noise occurrence. And after this, a reader may expect a brief overview of what has been done (or not done) in the past to evaluate the noise and what remains unexplored. More references and explanations are needed here. Otherwise, the purpose of this study does not seem sufficiently justified.

*Although the ESR facility like other IS radars is built with high gain and low noise performance owing to  its transmitted power (up to a maximum of 1.0 MW), antenna sensitivity (42 m*

*diameter) and high latitude location* $(78°09'11''N)$, *there are noise from other sources such as the signal-to-noise ratio (SNR) that varies inversely as the square of the distance from the receiver to the target (i.e., $S \propto R^{-2}$), noise associated with clutter in altitude up to 140 km (Wannberg et al., 1997) and the electromagnetic noise at the background.* Lehtinen (1989) and Vierinen et al., (2008) have suggested that the accuracy of the autocorrelation function in radar backscatter is limited as a result of disturbances from noise. David et al. (2018) worked on the technique to filter the real data from noise, but no statistical analysis to quantify the level of noise was carried out. Li et al. (2020) in their attempt to simulate the SNR of a proposed ISR (phased array radar) and compared with an equivalent parabolic dish radar, showed theoretically through their findings that the SNR from the phased array radar is weaker compared to that of the equivalent parabolic dish, whereas the analysis of noise and its error were left for future work.

In order to avoid radar data that are susceptible to clutter as a result of mountainous topography of Svalbard (David et al., 2018), *the data analysed in this work were observed by the EISCAT Svalbard Radar (ESR) 42 m dish between the altitude range of 100 and 470 km (where noise associated to clutter and background electromagnetic effect have been filtered) with a time resolution of 1 minute. As such, the focus of this paper is the analysis of the statistical occurrence of noise associated with different classes of ionospheric upflow, local time (LT) dependence, as well as* seasonal variability of the noise during ESR observations of upwelling ions at solar minimum of 2007 – 2008 shown in Figure 1.

*Such statistical studies have potential application in the improvement of the EISCAT instrumentation. For example, in the development of the upgrade of the existing EISCAT radars, the EISCAT 3D. This is because, for example, noise from sources such as the signal-to-noise ratio influence the temporal resolution of the EISCAT 3D radar measurements (Stamm et al., 2021). The EISCAT 3D radar relies on a high-power and phased array system can produce three-dimensional imaging of the upper atmospheric structures and processes in high resolution (McCrea et al., 2015). With such high-resolution imaging capabilities of the EISCAT 3D radar data, they can enhance research in, for instance, ionospheric electron densities and ion flow velocities. Thus, the present study can contribute to the development of the recent EISCAT 3D radar.*

The statement that noise is associated with non-physical velocities (ll. 43-44) hardly needs so many references. And they all seem rather formal, since the papers mentioned are actually in-depth studies of various aspects of radar observations, naturally using only physically meaningful values.

*The number of references has been reduced. The statement now reads:*

*Noise or rejected data in this study refers to ISR data with very high values of unphysical velocities above 10 km s$^{-1}$ unintentionally obtained during incoherent scatter analysis (Jones et al., 1988; Blelly et al., 1996; David et al., 2018).*

To avoid confusion and ambiguity, it would be much better to make the introduction as a separate section and add more relevant information there. The next section should be Instrumentation & data. The classification of fluxes should certainly be moved to this second section.

*The Introduction has been made a separate section, likewise Instrumentation. More relevant information has been added to the introduction as indicated in the preceding page.*

**Instrumentation and Data**

*The primary data used for this work is sourced from EISCAT Svalbard radar (ESR) during the international polar year (IPY) campaign in 2007.*

- *The ESR is a fixed and field-aligned 42m dish.*
- *Basic ionospheric parameters measured by the ESR are the electron density, electron and ion temperature and, the ion velocity which are respectively abbreviated as: $n_e$, $T_e$, $T_i$, and $v_i$*
- *About 300 days observation of 312,444 field-aligned profiles was made*
- *The observation occurs during a deep solar minimum as shown in Figure 1*

*The ESR observations of upwelling ions at solar minimum of 2007 – 2008 shown in Figure 1, indicates that the maximum daily total sunspot number is 66.0 in 2007 and 60.0 in 2008. Likewise, the maximum daily F10.7 radio flux over the same period as shown in Figure 1 is 93.9 and 88.6 in 2007 and 2008 respectively. Noise or rejected data in this study refers to ISR data with very high values of unphysical velocities above 10 km s[-1] unintentionally obtained during incoherent scatter analysis (Jones et al., 1988; Blelly et al., 1996; David et al., 2018). The classes of flux ($\geq 7.5 \times 10^{13} m^{-2} s^{-1}$; Wahlund & Opgenoorth, 1989) in this study and the filtering methodology follow the work by David et al. (2018), where upflows are categorised as follows:*

*Low-flux upflow:* $\quad 1.0 \times 10^{13}\ m^{-2}\ s^{-1} \leq f_{ion} < 2.5 \times 10^{13}\ m^{-2} s^{-1}$

*Medium-flux upflow:* $2.5 \times 10^{13}\ m^{-2}\ s^{-1} \leq f_{ion} < 7.5 \times 10^{13}\ m^{-2} s^{-1}$

*High-flux upflow:* $\quad f_{ion} \geq 7.5 \times 10^{13} m^{-2} s^{-1}$

Although section 2 is titled Results and Discussion, this reviewer did not find any discussion. Only the two last sentences can be considered somewhat related to the discussion. And it is too few. The discussion should be expanded, or the word "discussion" should be removed from the title. The results without discussion seem not a good idea though, especially if the introduction is too brief. There can be different ways to have an interesting discussion, e.g. implementation of the results obtained (for EISCAT 3-D?), their physical meaning, comparison with previous results.

The statement below has been added to the discussion.

*In the light of the above, the proposed phased array ISR, named Sanya ISR should take into cognisance, an ISR that in practice, will have a better SNR by ensuring the best input radar system constants, effectual scattering volume, and spatial variability terms in space, as stated in the work of Li et al. (2020). The results of this work could also be integrated in the buildup of the EISCAT 3D to allow for comparison in the SNR of the Scandinavian Arctic infrastructure and the Sanya ISR, which is proposed to be the first multistatic ISR in a low latitude region.*

Reply
**Citation**: https://doi.org/10.5194/angeo-2023-37-RC3

**New references**

Lehtinen, M.S., 1989. On optimization of incoherent scatter measurements. *Advances in Space Research*, *9*(5), pp.133-141.

Li, M., Yue, X., Zhao, B., Zhang, N., Wang, J., Zeng, L., Hao, H., Ding, F., Ning, B. and Wan, W., 2020. Simulation of the signal-to-noise ratio of Sanya incoherent scatter radar tristatic system. *IEEE Transactions on Geoscience and Remote Sensing*, *59*(4), pp.2982-2993.

Stamm, J., Vierinen, J., Urco, J.M., Gustavsson, B. and Chau, J.L., 2021, February. Radar imaging with EISCAT 3D. In *Annales Geophysicae* (Vol. 39, No. 1, pp. 119-134). Copernicus GmbH.

Vierinen, J., Lehtinen, M.S., Orispää, M. and Virtanen, I.I., 2008, September. Transmission code optimization method for incoherent scatter radar. In *Annales geophysicae* (Vol. 26, No. 9, pp. 2923-2927). Göttingen, Germany: Copernicus Publications.

---

## Author Response (AR2)

**Response - Technical Correction to the Accepted Version**

The authors deeply appreciate the reviewer for taking time to painstakingly go through the manuscript and for the valuable recommendations.

From Line 44 in the accepted manuscript

~~In order to avoid radar data that are susceptible to clutter as a result of mountainous topography of Svalbard (David et al., 2018), the data analysed in this work were observed by the EISCAT Svalbard Radar (ESR) 42 m dish between the altitude range of 100 km and 470 km (where noise associated to clutter and background electromagnetic effect have been filtered) with a time resolution of 1 minute. As such, the focus of this paper is the~~ The data has been filtered to avoid terrain clutter and background electromagnetic effects. The data used comes from the EISCAT Svalbard Radar (ESR) 42 m dish, using altitude ranges between 100 km and 470 km, and a time resolution of 1 minute. This will facilitate the goals of the present paper, which is the analysis of the statistical occurrence of noise associated with different classes of ionospheric upflow, local time (LT) dependence, as well as seasonal variability of the noise during ESR observations of upwelling ions at solar minimum of 2007 – 2008 shown in Figure 1, where the maximum daily total sunspot number is 66.0 in 2007 and 60.0 in 2008.